# An FPGA-Based Laser Virtual Scale Method for Structural Crack Measurement

**Miaomiao Yuan** [1], **Zhuneng Fang** [2], **Peng Xiao** [2], **Ruijin Tong** [3,*], **Min Zhang** [4] **and Yule Huang** [4]

1   Civil Engineering School, Guangzhou City University of Technology, Guangzhou 510800, China
2   Guangzhou Highway Engineering Group Co., Ltd., Guangzhou 510000, China
3   Guangzhou Nasha New Area Industrial Zone Development Authority, Guangzhou 511466, China
4   School of Civil Engineering and Transportation, South China University of Technology,
    Guangzhou 510640, China
*   Correspondence: t13512734366@163.com

**Abstract:** Real-time systems for measuring structural cracks are of great significance due to their computational and cost efficacy, inherent hazards, and detection discrepancies associated with the manual visual assessment of structures. The precision and effectiveness of image measurement approaches increased their applications in vast regions. This article proposes a field-programmable gate array (FPGA)-based laser virtual scale algorithm for noncontact real-time measurement of structural crack images. The device first sends two parallel beams and then applies image processing techniques, including de-noising with median and morphological filtering, as well as Sobel-operator-based edge extraction, to process and localize the light spots. Afterwards, it acquires the scale of the pixel distance to the physical distance and then derives the actual size of the crack. By processing and positioning, the FPGA acquires the scale of the pixel distance to the physical space and then derives the actual size of the crack. The experimental study on crack measurements demonstrates that the proposed technique has precise and reliable results. The error rate is approximately 2.47%, sufficient to meet measurement accuracy criteria. Moreover, experimental results suggest that the processing time for one frame using an FPGA is about 54 ms, and that the hardware acceleration provided using an FPGA is approximately 120 times that of a PC, allowing for real-time operation. The proposed method is a simple and computationally efficient tool with better efficacy for noncontact measurements.

**Keywords:** FPGA; crack detection; laser virtual scales; noncontact measurement

## 1. Introduction

A wide variety of techniques have been proposed for nondestructive crack and surface flaw detection in structures, such as radiography [1], ultrasonic test [2], magnetic particles [3], acoustic emission [4], and image processing [5].

Noncontact measurement techniques based on image processing are becoming the mainstream methods for structural identification and monitoring [6,7]. Machine vision is widely applied in crack detection [5], spalling detection [8], bolt loose detection [9], load estimation [10], and strain monitoring [11]. Existing crack image detection techniques generally use hardware devices to capture images and then transmit them to back-end software for processing. When large-span bridges and high-rise structures are subjected to mobile scanning inspection, image detection techniques cannot meet the demand for real-time crack analysis, which involves processing a large number of images within a short time. The real-time health monitoring is still a challenge for these existing approaches. Therefore, there is an urgent need to further develop new technologies that operate in real time and can accurately detect cracks and measure their width automatically [12]. Field-programmable gate arrays (FPGAs) are now widely applied in real-time image processing due to their fast image processing speed and other advantages [13,14]. In comparison with

computer-aided monitoring systems, the FPGA-based technology increases the speed and reduces the energy consumption [14].

In image processing techniques, several algorithms such as image filtering, noise reduction, edge detection, and image restoration are often used to process crack images [15]. Research on crack image filtering and noise reduction includes traditional mean filtering [16], median filtering [17], adaptive filtering [18], and noise reduction algorithms [19]. Research on crack edge detection includes Roberts [20], Sobel [21], Prewitt [22], and Canny [23] operators. Research on image restoration includes morphological filtering [24], direct regularized restoration [25], and iterative approaches [26].

In traditional image measurement techniques, camera calibration algorithms are often utilized for mapping the world coordinate system to the pixel coordinate system [27]. However, the calibration process requires a multitude of calibration plate photographs, and for most measurement environments, there are often many limitations in placing calibration plates. To address practical application limitations, optical measurement techniques are widely used in the field of noncontact measurement, for example, obtaining the image scale using a laser rangefinder [28] or a laser virtual scale [29]. Although many papers have studied the classification and detection of a crack's presence through images [30] and vibration data [31], the literature still lacks a reliable method for the determination of the crack width. This study contributes to addressing this issue, and it would be beneficial to combine the method used in the paper with those algorithms.

Moreover, compared with traditional GPU and CPU hardware image processing chips, FPGAs offer fast image processing as well as low cost and low power consumption, which are advantageous in real-time crack measurements. Using FPGAs for image processing, involving filtering and noise reduction [32], edge extraction [33], image enhancement [34], image calibration [35], and image localization [36] and other related algorithm processing, the computational efficiency can be improved considerably.

This paper presents a laser virtual scale method based on fast FPGA image processing to enhance the applicability of noncontact evaluation through image measurements. Real-time crack measurement is implemented, having features of low cost and power consumption with better precision. In the proposed method, there is no need to place a calibration plate on the structure in crack measurement. This approach makes the noncontact measurement more practical; compared with a complex camera calibration matrix transformation algorithm, the spot localization and centroid extraction used in this method are more favorable to FPGA implementation in terms of algorithm logic [37]. With the fast FPGA data processing, there is no need to fit the full-field scale to the image [38] to obtain the scale of each measurement object in the image. The cracks can be detected one by one in real time according to the local image scale.

The remainder of this paper Is organized as follows. Section 2 demonstrates the laser virtual scale algorithm used during the image acquisition. Section 3 presents the FPGA processing technique to improve the image with laser spots. Section 4 further presents laser spot measurement and crack measurement using the FPGA technique. A verification example is presented in Section 5. The conclusions of this study are drawn in Section 6.

## 2. Laser Virtual Scale Model

A diagram of the laser virtual scale is shown in Figure 1. This device uses the distance between two parallel laser beams emitted from an optical beam-splitting prism as an absolute reference, since this distance is fixed. Thus, any target can be measured by comparison with this fixed spot distance. To this end, the first step is to find the scale between the fixed spot distance and image pixel distance. Subsequently, the real size of the target can be obtained via the scale and pixel number, which is discussed in the following.

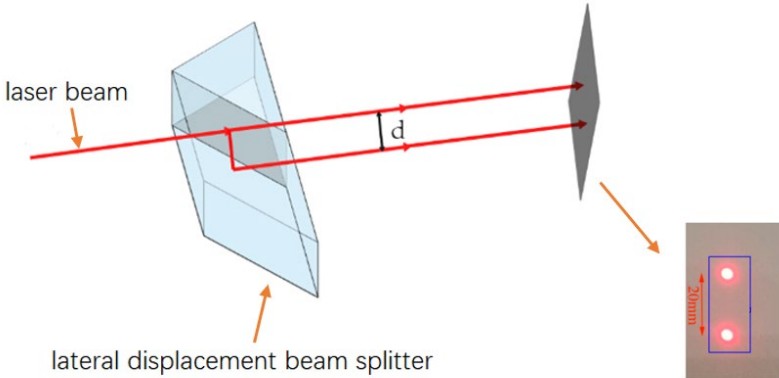

**Figure 1.** Theoretical diagram of the laser virtual scale.

The semiconductor laser (1mw, 635nm, laser diode module for laboratory use; Edmund Optics (Shenzhen) Co., Ltd., Shenzhen, China) selected in this study has a small beam divergence angle and sufficient output power to ensure that the collected spot size is small and clear; the beam-splitting prism has a beam-splitting interval of 10.00 mm and a parallelism of less than 30 arcsec, which ensures that the beam spacing is almost fixed.

As shown in Figure 2, assume that there is a length $ds$ in the world coordinate system and that the inclination angle of $ds$ is measured using an inclinometer attached to the spectroscopic prism. The pixel distance of $ds$ in the pixel coordinate system is denoted as $ds'$.

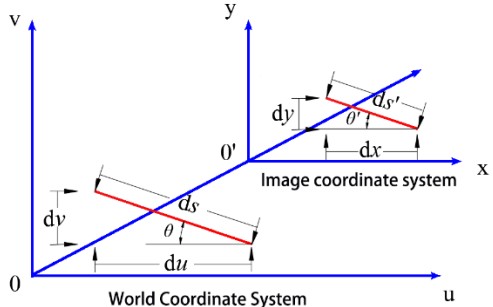

**Figure 2.** Coordinate system mapping.

Thus, the horizontal and vertical projections of $ds$ and $ds'$ are given by Equations (1) and (2), respectively:

$$du = ds \cdot \cos\theta; dv = ds \cdot \sin\theta \tag{1}$$

$$dx = ds' \cdot \cos\theta'; dy = ds' \cdot \sin\theta' \tag{2}$$

where $\theta' = \arctan(\frac{y_1 - y_0}{x_1 - x_0})$ and $(x_0, y_0)$ and $(x_1, y_1)$ are the coordinates of the two ends of $ds'$ in the pixel coordinate system.

The scale relationship is expressed as

$$du = dx \cdot h(x, y); dv = dy \cdot s(x, y) \tag{3}$$

where $(x, y)$ is the centroid of $ds'$, $h(x, y)$ is the local horizontal scale of the midpoint of $ds'$, and $s(x, y)$ is the local vertical scale of the midpoint of $ds'$. The expression can be given as

$$(x, y) = \left( \frac{x_0 + x_1}{2}, \frac{y_0 + y_1}{2} \right) \tag{4}$$

$$h(x, y) = \frac{du}{dx} = \frac{ds \cdot \cos\theta}{|x_1 - x_0|}; s(x, y) = \frac{dv}{dy} = \frac{ds \cdot \sin\theta}{|y_1 - y_0|} \tag{5}$$

The laser virtual scale device presented in this paper uses spot centroids to determine the pixel distances between the two parallel beams, and $(x_0, y_0)$ and $(x_1, y_1)$ are the coordinates of the centroids in the pixel coordinate system.

### 3. FPGA-Based Laser Spot Image Processing

Before positioning the spot and extracting the centroid, the spot image must be processed. For real-time acquisition, the shooting environment is usually not favorable, and poor imaging quality is often caused by factors such as poor lighting conditions and shooting difficulties, making it difficult to extract effective laser spots for scale measurement. At the same time, because of substantial impurities, dust, and an uneven surface around the measurement target, the acquired laser spot image contains considerable noise interference that severely interferes with the positioning of the laser spot [39].

*3.1. Image Processing*

(1)　Preliminary de-noising

Pepper noise is the most common form of noise in an image measurement process, and the key to processing a laser spot image is pepper noise filtering and edge extraction [40]. Median filtering has a favorable effect on pepper noise filtering and moreover does not blur the crack edge information; therefore, median filtering is employed for initial de-noising of the laser spot image [41].

Median filtering involves a nonlinear smoothing count based on ranking statistical theory, characterized by choosing a suitable filter template and applying that template as a sliding window to use the median value of the pixel's field as a new gray value for that pixel. The specific expression is

$$g(i, j) = med\{g(m, n)\} \tag{6}$$

where $g(i, j)$ is the new gray value of the pixel after median filtering and $g(m, n)$ is the gray value of each pixel within the filter template.

(2)　Local adaptive edge extraction

In this study, the Sobel operator is used to convolve the spot image to extract the spot edges, and the Sobel convolution factors corresponding to the x and y directions are shown in Figure 3.

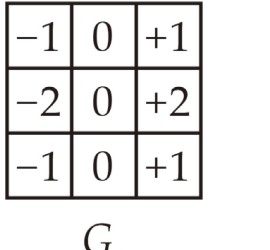 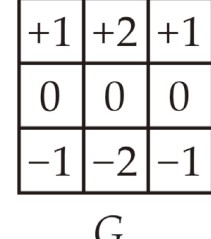

**Figure 3.** Sobel convolution factor.

Assuming that *A* is the original crack image, the expression for the gradient amplitudes corresponding to the *x* and *y* directions (*Gx and Gy*) are as follows:

$$Gx = \begin{bmatrix} -1 & 0 & 1 \\ -2 & 0 & 2 \\ -1 & 0 & 1 \end{bmatrix} \times A, Gy = \begin{bmatrix} 1 & 2 & 1 \\ 0 & 0 & 0 \\ -1 & -2 & -1 \end{bmatrix} \times A \tag{7}$$

Therefore, the adaptive Sobel edge detection algorithm can be divided into the following four steps: (1) calculate the product of the Sobel convolution factor and each line of a $3 \times 3$ window template; (2) find *Gx* and *Gy* after the operation with the $3 \times 3$ window

template; (3) find the value of the gradient amplitude $G = \sqrt{G_x{}^2 + G_y{}^2}$; and (4) use the median of the gray values of the pixel points in the $3 \times 3$ window as a local adaptive threshold [42]. By using the adaptive threshold setting, the edge detection threshold of each region within the crack image changes with the image features of that region.

(3) Deep de-noising

The noise is extracted as "edge points" in the process of crack extraction because of the prominent difference in gray values between the noise and the surrounding pixels, which makes it difficult to completely eliminate the noise. Therefore, it is necessary to continue removing noise from the spot image. Morphological filtering can effectively eliminate the small noise remaining after the initial de-noising and edge extraction. This filtering is divided into morphological erosion and morphological expansion algorithms [43].

The expression for morphological expansion is shown in Equation (8), and the expression for morphological corrosion is shown in Equation (9).

$$g(x, y) = max(I(x + i, y + i)) \quad (i, j) \in D_b \tag{8}$$

$$g(x, y) = \min(I(x + i, y + i)) \quad (i, j) \in D_b \tag{9}$$

$$D_b = \{(i, j) | -\gamma \leq i \leq \gamma, -\gamma \leq j \leq \gamma\} \tag{10}$$

where $D_b$ denotes the area covered by the convolution kernel, $g(x, y)$ denotes the pixel value after morphological operations, and $I(x, y)$ denotes the image pixel value at each point in the area.

For morphological filtering algorithms, there are usually two methods, i.e., open and closed operations [44]. An open operation is an erosion operation followed by an expansion operation, while a closed operation is an expansion operation followed by an erosion operation. Among them, an open operation can effectively remove isolated noise, burrs, and other fine objects while maintaining the position and shape of the spot, and using an open operation can achieve the purpose of deep de-noising.

### 3.2. FPGA Hardware Implementation

The key to the above laser spot image processing algorithm using an FPGA is the extraction of the sliding window [45]. In this study, we used a Shift-RAM shift register to implement extraction of the window template. As shown in Figure 4, a Shift-RAM shift register works as follows: set the number of lines and the width of each line, and when the next data input comes in, the previous data input goes forward; when the data input reaches the width of a line, the first data input goes into the next line shift storage, and so on, until the data window is filled. Quartus II software (13.1, Intel Corporation, Santa Clara, CA, USA) has its own Shift-RAM IP cores, which call two Shift-RAM IP cores to shift and store two lines of data to form a data window with the third line of newly entered data. The use of Shift-RAM IP cores in Quartus II requires that the parameters be set according to the window size.

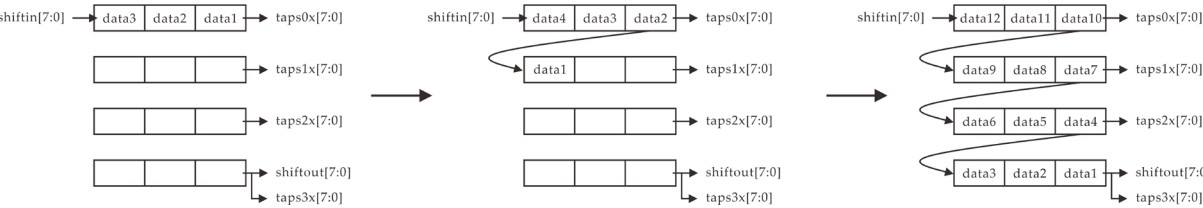

**Figure 4.** Shift register operation principle.

The window extraction hardware block diagram is shown in Figure 5.

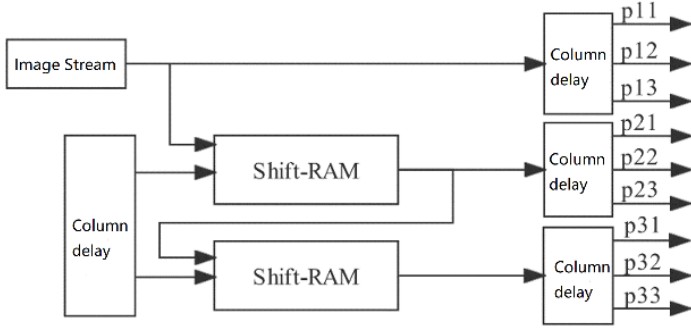

**Figure 5.** Block diagram of the 3 × 3 window extraction window.

After extracting the 3 × 3 sliding window, the FPGA-based laser spot image processing algorithm is implemented using the Verilog hardware description language according to the relevant formula in Section 3.1. The RTL generated after synthesis of the program is shown in Figure 6.

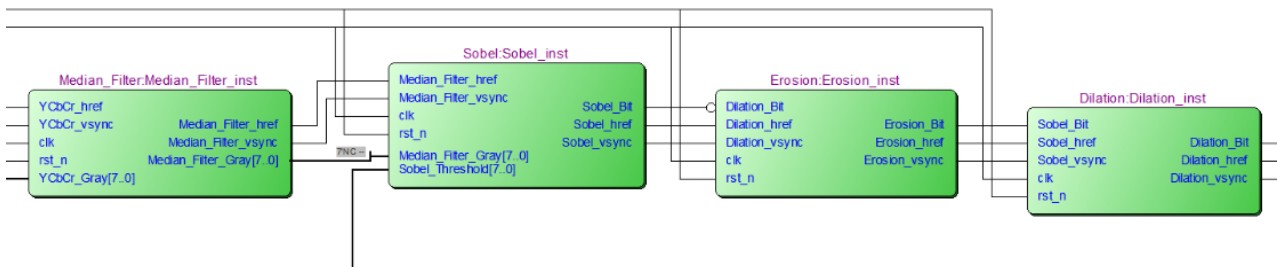

**Figure 6.** FPGA-based image processing algorithm RTL diagram.

A joint simulation is performed using Quartus II and ModelSim (2020.4, Siemens Digital Industries Software, Plano, TX, USA). Following the simulation, the simulation waveforms of the module can be viewed as shown in Figures 7 and 8. As shown by the blue block diagram in Figure 9, the results of the signal simulation data exemplified in the block diagram match those of the relevant image processing algorithm; thus, the module is verified via simulation.

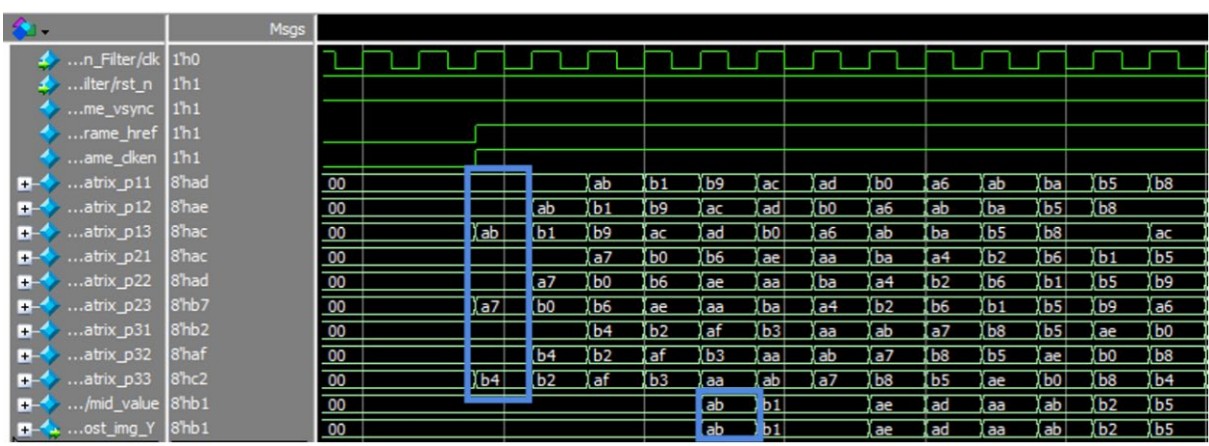

**Figure 7.** Median filtering algorithm simulation module.

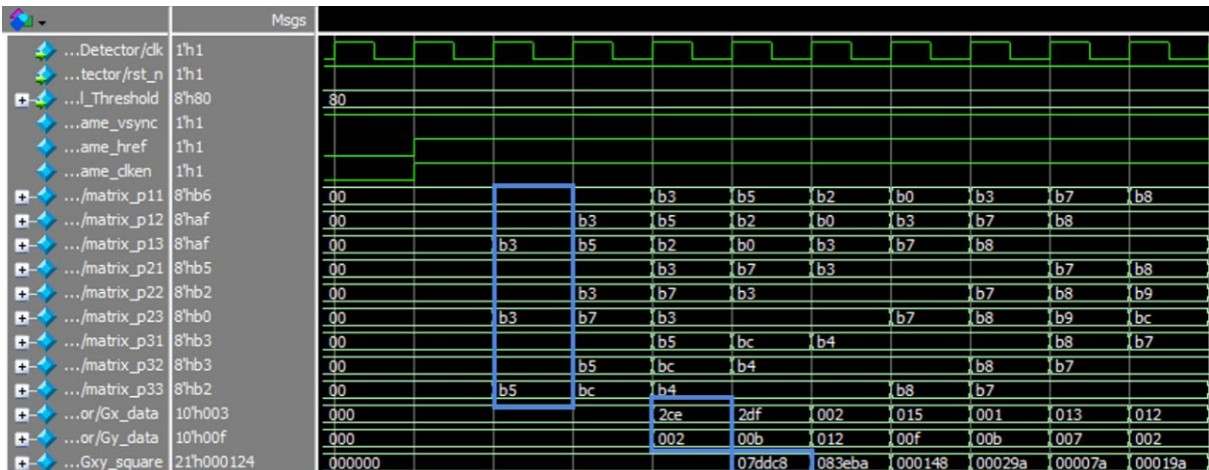

**Figure 8.** Sobel edge detection algorithm simulation module.

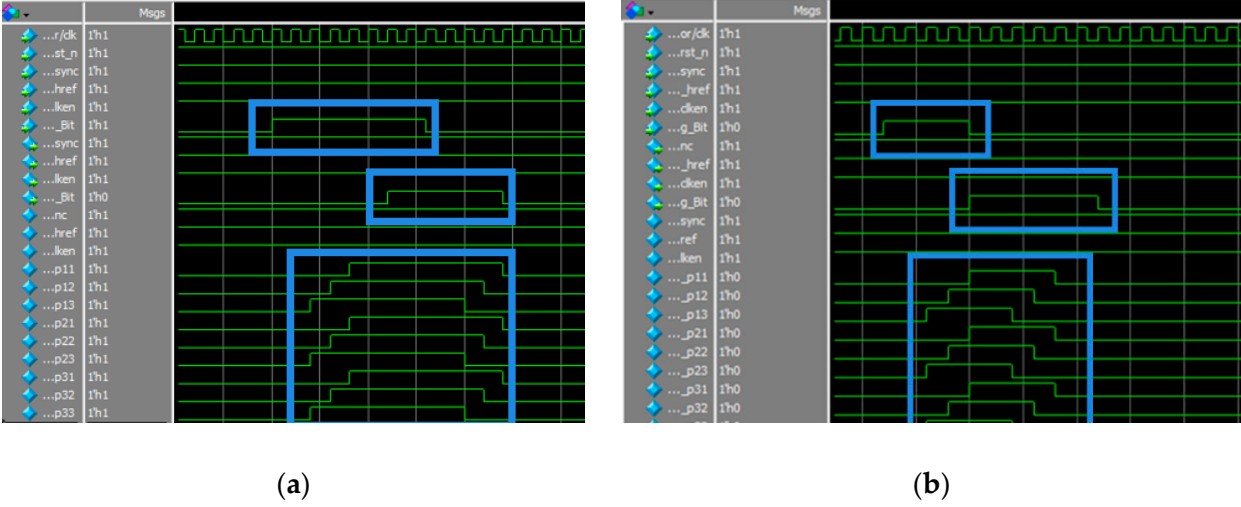

(**a**)

(**b**)

**Figure 9.** Morphological filtering algorithm simulation module. (**a**) Morphological expansion. (**b**) Morphological corrosion.

## 4. FPGA-Based Spot Localization and Centroid Extraction

The pixel distances of the spots are of great importance for the measurement. However, the image spot center is not easy to localize, especially when the edge of the spot is a blur. This section presents FPGA-based spot localization and center extraction algorithms.

### 4.1. Spot Localization and Plasmonic Extraction

The spot images after filtering, noise reduction, and edge extraction are localized using the connected domain markers. A set of pixel points consisting of valid pixel points at the edges of adjacent spots is found, and each spot is given its own marker. The purpose of spot marking is to distinguish different spots in the image so that their coordinate information can be counted.

The specific process of FPGA-based spot marking is as follows:

(1)  Obtain a pixel point scan of the spot edge image by row. When an unmarked valid pixel $I(x, y)$ is scanned and the pixel point markers in all eight fields of the pixel are 0, then a new marker is given to $I(x, y)$. Continue scanning the row to the right, and if the pixel point to the right of $I(x, y)$ is an unmarked valid pixel point, then assign the same marker to the pixel point to the right of $I(x, y)$.

(2)  While scanning the current row, the next row is marked, i.e., the valid pixel point of the marked pixel in the field of the next row is marked and given the same marker number as the valid pixel point. By scanning row by row, the adjacent pixel points are marked with the same marker.

(3)  While scanning the image, a storage space is opened to store the coordinate data and address the information of each connected domain, and data statistics are performed to calculate the number of points and centroid information inside the connected domain.

(4)  When the pixel below the neighbor of the marked pixel is marked in the same connected domain, the marking of the connected domain is considered completed, and the statistical results as well as the centroid information are output.

*4.2. FPGA Hardware Implementation*

The FPGA-based hardware module connection, as shown in Figure 10, includes the following four modules: a scan-and-store module, marker module, evaluation module, and FIFO cache module.

First, the 2 × 2 window is generated using the Shift-RAM IP core. The scan module uses the 2 × 2 window to scan the input image pixels and outputs valid pixels as well as the address information (*addr_wr)* to the marker module.

The labeling module labels the valid pixels according to the process described in the previous section and passes the statistical address information (*addr_labe) l* to the scanning module.

If the pixel below the neighboring domain is determined to be the same connected domain marker, the evaluation module determines that the connected domain scanning is completed. At this time, the connected domain scanning completion enables signal (*done_en)* to be raised for one clock cycle, and the connected domain information (*data_out)* is output to the evaluation module. This module calculates the connectivity data information and outputs the result. Only the coordinates of the centroid are presented in this paper.

The FIFO cache module can play a temporary data storage function and can make the subsequent processing flow smoothly, which can prevent the back stage processing too late in the event of a front stage burst, leading to data discarding.

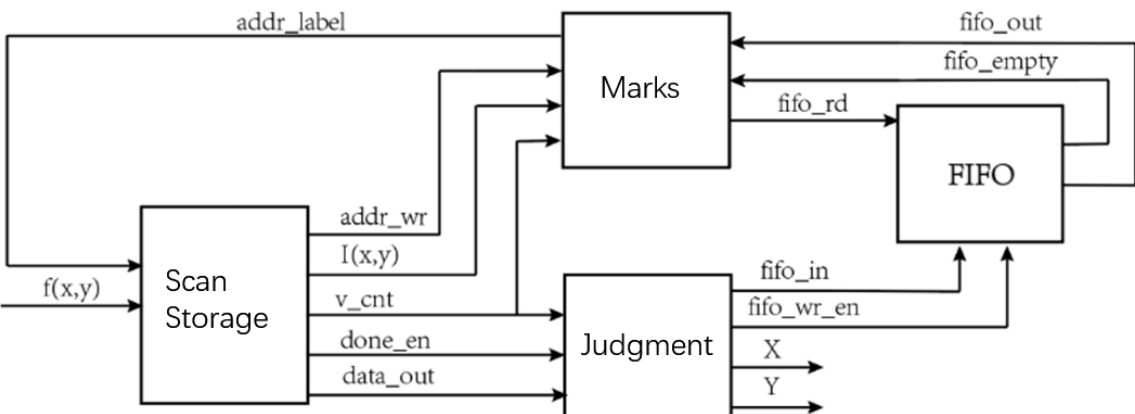

**Figure 10.** FPGA hardware implementation.

**5. Experimental Verification**

The purpose of this experiment was to measure crack width using the laser virtual scale algorithm. The main pieces of equipment are a semiconductor laser emitting the beam, a camera to capture the image, and an FPGA for image processing. Figure 11 shows the system diagram of the hardware device for this experiment, using the FPGA development board, the laser virtual scale to measure the crack images, and a laptop computer to simulate

and verify the laser virtual scale algorithm. For simplicity, the laser virtual scale device projects a parallel spot with vertical spacing toward the target crack.

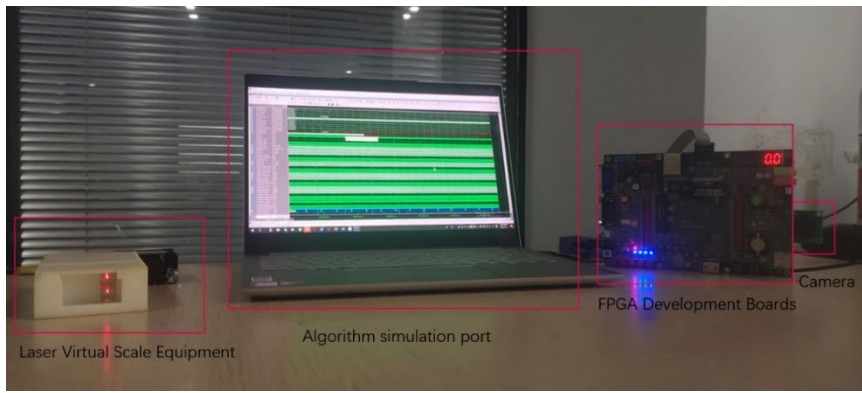

**Figure 11.** Crack measurement experimental setup.

*5.1. Accuracy Verification*

Figure 12 shows the raw crack map as well as the results of filtering, noise reduction, and edge detection of the crack using the FPGA. It should be noted that there is no restriction on the direction of a crack.

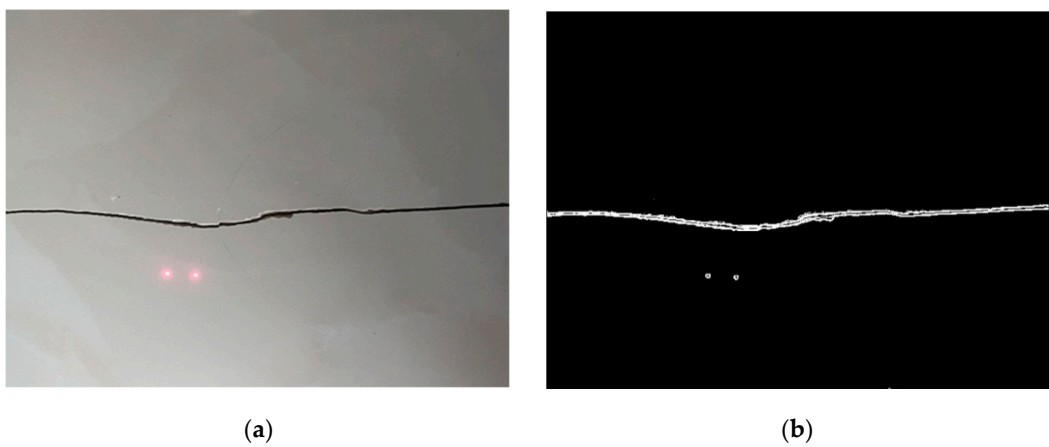

(**a**)                                                            (**b**)

**Figure 12.** Crack width measurement experiment. (**a**) Original picture of the crack experiment. (**b**) Crack map after image processing.

To verify the accuracy of the laser virtual scale algorithm implemented on an FPGA, a simulation program was developed in the Verilog language to simulate and verify the experimental map of the spot taken by the camera. The centroid information from the simulation is displayed in Figure 13.

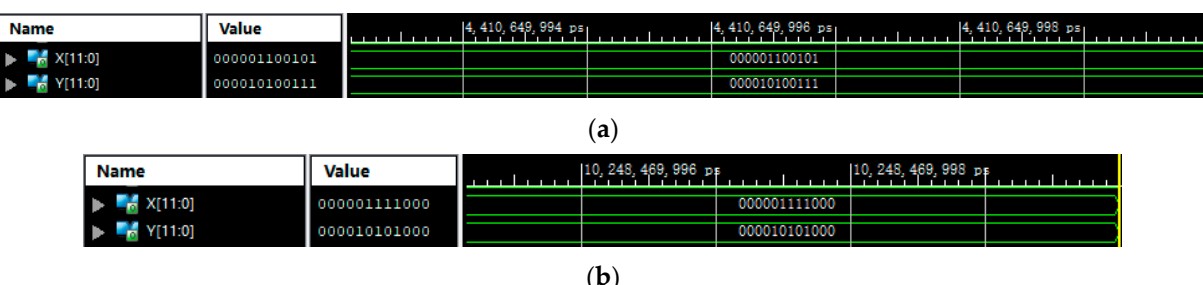

**Figure 13.** Simulation results of centroid coordinates. (**a**) Left spot centroid coordinates. (**b**) Right spot centroid coordinates.

The simulation and MATLAB (2020b, MathWorks, Natick, MA, USA) centroid calculation results are compared in Table 1.

**Table 1.** Comparison table of the FPGA simulation and MATLAB calculation results.

|  | Left Coordinate | Right Coordinate | Barycenter Distance (Pixels) |
|---|---|---|---|
| FPGA simulation | (203, 335) | (241, 336) | 38.013 |
| MATLAB | (205, 336) | (242, 338) | 37.054 |

The comparison indicates an error between the FPGA simulation results and the MATLAB calculated transverse and longitudinal coordinates of the centroid. Assuming that the MATLAB calculation is accurate, the measurement error of the centroid distance is 2.6%, and the error of the spot spacing is 0.26 mm for a fixed 10 mm.

The actual width $D$ of the crack is obtained using the above FPGA centroid extraction with a pixel distance of $ds$, a spot fixed pitch $DS$ of 10 mm, and a maximum pixel width $D$ of the crack:

$$D = \frac{10}{d_s} \cdot d \tag{11}$$

As shown in Figure 14, the maximum pixel spacing of the crack in the vertical direction obtained via simulation is 6 pixels, and the crack width $D$ measured in this experiment is 1.58 mm according to Equation (11).

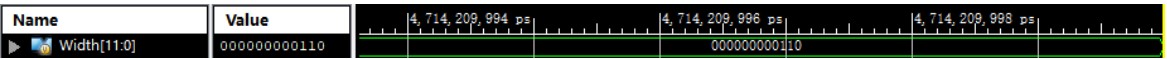

**Figure 14.** Crack maximum pixel pitch simulation results.

Contact crack measurement was carried out using a ZBL-F120 crack width gauge (ZBL-F120 Crack measurement, Zhilianbo Company, Beijing, China), and the measurement experimental and result graphs are shown in Figure 15.

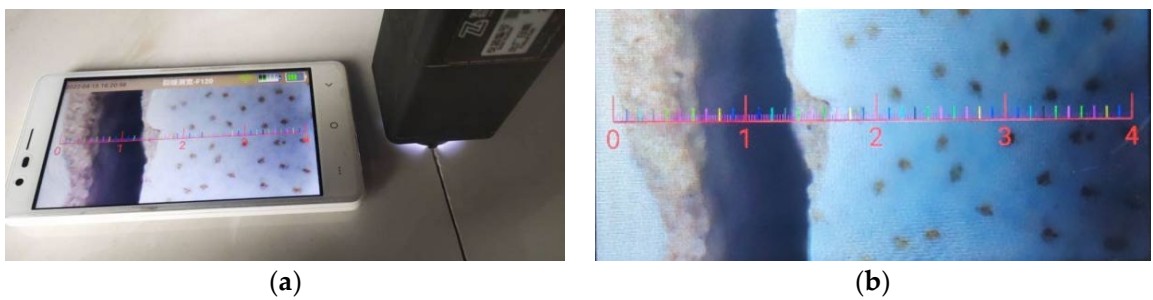

(**a**)          (**b**)

**Figure 15.** Experimental diagram of the crack with gauge measurements: (**a**) experimental operation graph; (**b**) experimental results graph.

The figure shows that the actual maximum width of the crack measured using the crack gauge is 1.62 mm. This experiment uses an inclinometer to adjust the beam projection angle to 90°, 60°, and 45° to verify the effect of the projection angle on the measurement results. The error analysis is presented in Table 2.

**Table 2.** Error analysis table under different beam projection angles.

| Laser Angle | 90° | 60° | 45° |
|---|---|---|---|
| Actual width | 1.62 mm | 1.62 mm | 1.62 mm |
| Calculated width | 1.58 mm | 1.50 mm | 1.42 mm |
| Error | 2.47% | 7.4% | 12.3% |

The errors are 2.47%, 7.4%, and 12.3% when the laser virtual scale equipment beam projection is 90°, 60°, and 45° to the target plane, respectively. Therefore, when using the laser virtual scale equipment, the beam should be controlled close to the vertical projection to ensure measurement accuracy. However, in terms of direction of the crack there is not any limitation.

*5.2. Resource Consumption and Real-Time Analysis*

(1) Resource consumption

The resources utilized by this system to implement the FPGA-based laser virtual scale algorithm are shown in Table 3. In this system, the slice (logic cell) occupies 23%, one slice contains four lookup tables (LUTs) and eight registers inside, and the proportions of LUTs and registers are 16.18% and 7.81%, respectively, because in actual use, not all LUTs and registers inside each slice are fully occupied. There are 116 B-RAM/FIFO units (memory modules) in the chip, 22 of which are actually used with a utilization rate of 18.97%. The system needs to communicate with external devices, so it takes up part of the IO resources, with a utilization rate of 18.98%.

**Table 3.** Device utilization summary.

| Logic Utilization | Used | Available | Utilization |
|---|---|---|---|
| Number of slice registers | 4275 | 54,576 | 7% |
| Number of slice LUTs | 4497 | 27,288 | 16% |
| Number of fully used LUT-FF pairs | 1652 | 7120 | 23% |
| Number of bonded IOBs | 60 | 316 | 18% |
| Number of block RAM/FIFO | 22 | 116 | 18% |
| Number of BUFG/BUFGCTRLs | 2 | 16 | 12% |
| Number of DSP48A1s | 2 | 58 | 3% |

(2) Real-time analysis

To measure the results in real time, this system saves the image pair in DDR2 and then processes the image 1000 times repeatedly to measure the overall time, thus obtaining the average time required to process each frame and calculating the average processing speed. In addition, the laser virtual scale algorithm was implemented on the laptop side using VS2017 and OPENCV, and the running speeds of both are shown in Table 4.

**Table 4.** Speed table of the laser virtual scale algorithm running on an FPGA and a PC.

| Platform | Image Resolution | Time Spent Per Frame | Frame Rate |
|---|---|---|---|
| FPGA | 640 × 480 | 54 ms | 18.52 fps |
| PC | 640 × 480 | 6570 ms | 0.15 fps |

On average, it takes 6570 ms (0.15 fps) for a PC to process each frame to obtain the laser virtual scale ratio, while it takes approximately 54 ms (18.52 fps) on the FPGA, which basically meets the real-time requirement. The hardware acceleration using the FPGA is approximately 120 times faster than a that using the PC.

**6. Conclusions**

In this paper, we propose an FPGA-based laser virtual scale method for noncontact real-time measurement of structural crack images. The laser virtual scale device emits parallel beams with fixed spacing, and the FPGA processes the spot image to remove noise in depth and then locates and extracts the centroid of the spot from the processed spot image to obtain the scale of the pixel distance to the physical distance for calculating the actual size of the crack. The lateral displacement beam splitter not only provides a fixed beam distance, but also the same spot shape, which guarantees that the centroids

of the spots are the same and ensures the precise measurement of the pixel distance. The crack measurement experiments show that the measurement error of the FPGA-based laser virtual scale is approximately 2.47%, which can meet the measurement accuracy requirements. Moreover, experimental analysis shows that the processing time of one frame on an FPGA is approximately 54 ms, and that the hardware acceleration achieved using an FPGA is approximately 120 times that of a PC, which can meet the real-time requirement.

This paper provides a simple and efficient method for structural crack measurement, which can be widely used in noncontact image measurement, civil engineering, etc. To this end, the device can be further integrated in a photo and video camera. A video consists of a sequence of images captured at a given frequency, and a single image can be obtained by stopping the video at any time during this sequence. Although the method was originally developed for images, it is applicable to videos.

The identification of microcracks is not aimed as the application of this device and can therefore be considered a further development of this research. In order to apply the proposed method in structural cracks, an operator is required to calibrate the camera toward the cracks. Considering the available datasets of crack images, this task can be performed through a machine learning algorithm trained on crack images. For instance, a classifier network can assist the operator in identifying the location of the cracks. In addition, the de-noising process can be assigned to a neural network to automatically extract a distinguishable map of the image. However, machine learning methods require a comprehensive set of image datasets for training, and the obtained models are commonly hard to interpret. On the other hand, the proposed method does not require an image dataset for training, and the working process is rather interpretable. Implementing this technology on automatic systems such as drones and robots can address its limitations in obstructed views and further boost the accuracy as well as time and cost efficiency. Moreover, the parameters affecting the accuracy of the method, such as the resolution of the image, distance between camera and crack, and lighting conditions, will be investigated in the next phase of this study.

**Author Contributions:** Conceptualization, M.Y.; methodology, M.Y.; software, M.Z.; validation, M.Y., M.Z. and Y.H.; formal analysis, Z.F.; investigation, M.Z. and Y.H.; resources, R.T.; writing—original draft preparation, Z.F. and P.X.; writing—review and editing, M.Y.; visualization, P.X.; supervision, R.T.; funding acquisition, M.Y. All authors have read and agreed to the published version of the manuscript.

**Funding:** This research was funded by [the Hundred-Talent Program of Guangzhou City University of Technology] grant number [YB180002] and The APC was funded by [Miaomiao Yuan].

**Institutional Review Board Statement:** Not applicable.

**Informed Consent Statement:** Not applicable.

**Data Availability Statement:** The data that support the findings of this study are available from the corresponding author upon reasonable request.

**Acknowledgments:** This paper was supported by the Hundred-Talent Program of Guangzhou City University of Technology (YB180002).

**Conflicts of Interest:** The authors declare no conflict of interest.

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
