# Peer review of "An FPGA-Based Laser Virtual Scale Method for Structural Crack Measurement"

_buildings, doi:10.3390/buildings13010261_

Round 1

Reviewer 1 Report

"An FPGA-based laser virtual scale method for structural cracks measurement" The article is interesting. A few observations are given below,

1) The abstract is not clear. An abstract is a short summary of your completed research. It is intended to describe your work without going into great detail. Abstracts should be self-contained and concise, explaining your work as briefly and clearly as possible. 

The presented abstract is too short.  The abstract should provide an overview of the proposed methods/methodology and materials with obtained results in the form of quantitative values. 

2) For readers to quickly catch the contribution of this work, it would be better to highlight major difficulties and challenges, and the authors' original achievements to overcome them, in a clearer way in the Introduction section. Also, the recent literature available in this field should be included in the introduction section. Highlight your objectives in the last paragraph of the introduction chapter. 

3) Equations must be quoted with proper references.

4) Moreover, the results and discussion are not clearly dealt with the outcomes of the proposed work. The authors should explicitly state the novel contribution of this work, and the similarities, and differences between this work with the previous publications in this section.

Reviewer 2 Report

This manuscript introduces an FPGA (Field Programmable Gate Array)-based laser virtual scale algorithm for noncontact real-time measurement of structural crack images is proposed in this study. In summary, the research is interesting and provides valuable results, but the current document has several weaknesses that must be strengthened in order to obtain a documentary result that is equal to the value of the publication.

General considerations:

(1)At the thematic level, the proposal provides a very interesting vision, as the automation of damage detection would be a very useful resource for engineers. However, crack detection is not limited to visual techniques. This issue is an important limitation about the aspirations of the proposal, whose limitations should be assumed with more rigour and realism in the development of the argumentation of the manuscript.

(2)The document contains a total of 32 employed references, of which 6 are publications produced in the last 5 years (19%), 5 in the last 5-10 years (16%), 10 than 10 years old (31%) and 11 undated (34%), implying a total percentage of 35 % recent references. In this way, the total amount is not enough and convincing is not high. More literatures of recent ten years need to be referred to.

(3)Technique concerns: Is it innovative and proven advanced in methods (how does it compare with other state-of-art methods)? What is the workload and feasibility/prospects?

Title, Abstract and Keywords:

(4)The abstract is complete and well-structured and explains the contents of the document very well. Nonetheless, the part relating to the results could provide numerical indicators obtained in the research.

Chapter 1: Introduction

(5)The first paragraph introducing the research topic may present a much broad and comprehensive view of the problems related to your topic with citations to authority references (https://doi.org/10.1016/j.engstruct.2022.115158 ). 

(6)The novelty of the study is not apparent enough. In the introduction section, please highlight the contribution of your work by placing it in context with the work that has done previously in the same domain.

(7)On a general level, the study of the proposed detection techniques is reasonable, and the explanation of the objectives of the work may be valid. However, the limitations of your work are not rigorously assumed and justified. 

(8)Vision technology applications should also be introduced for a full glance of the scope of related areas. For strain detection, please refer to seismic performance evaluation of recycled aggregate concrete-filled steel tubular columns with field strain detected via a novel mark-free vision method.

Chapter 2: Laser scale model

(9)Based on the complexity of the contents developed in chapter 3, it is noted that the scheme in figure 1 could be even more complex and detailed in the explanation of the processes.Like how two parallel laser beams are emitted.

(10)The type of laser used in this paper can be proposed..

Chapter 3: FPGA-based laser spot image processing

(11)I did not see the effect drawing of the image processing method proposed in 3.1.

Chapter 4: FPGA-based spot localization and centroid extraction

(12)The parameters in Figure 10 need to be detailed.

Chapter 5: Experimental verification

(13)It is necessary to explain the equipment used in the experiment, the type of instrument and the experimental environment in detail.

Chapter 6: Conclusion

(14)It should mention the scope for further research as well as the implications/application of the study.

(15)I recommend including the limitations regarding the consideration of damage indicated in this review in the limitations assessment. This part of the document can be improved and completed with more rigour.

Reviewer 3 Report

1. Can the method proposed in this paper be applied to video recording? In the field, data is acquired through drones or video. Is data analysis possible for video?

2. How does the method proposed in this paper work for micro cracks?

3. In Section 5.1, we analyzed accuracy. There can be errors by analysis and there can be errors by measurements. What about the analysis of each?

4. In section 5.1 the accuracy will depend on the distance between the camera and the crack. Resolution greatly affects accuracy. Have you considered the impact of this?

5. Is the directionality only possible in the transverse direction in crack measurement? Please specify in the paper if measurements can be made for other directions, such as depth.

6. Recently, studies to classify various damages as well as cracks in infrastructures are being conducted. In such research, algorithms such as deep learning and machine learning automatically classify cracks. What is a novel thing about this paper compared to these papers? The following papers should be reviewed more in-depth in the introduction to literature.

https://doi.org/10.1109/ICIP.2006.313007

https://doi.org/10.1080/10298436.2021.1945056

Round 2

Reviewer 2 Report

ACCEPT

Reviewer 3 Report

The authors addressed all comments from the reviewers.